Journal of
open psychology data

# A Dataset of 108 Novel Noun-Noun Compound Words with Active and Passive Interpretation

DATA PAPER

PHOEBE CHEN [iD]

DAVID POEPPEL [iD]

ARIANNA ZUANAZZI [iD]

*Author affiliations can be found in the back matter of this article

]u[ ubiquity press

## ABSTRACT

We created a dataset of 205 English novel noun-noun compounds (NNCs, e.g., "doctor charity") by combining nouns with higher and lower agentivity (i.e., the probability of being an agent in a sentence). We collected active and passive interpretations of NNCs from a group of 58 English native speakers. We then measured interpretation time differences between NNCs with active and passive interpretations (i.e., 108 NNCs), using data obtained from a group of 68 English native speakers. Data were collected online using crowdsourcing platforms (SONA and Prolific). The datasets are available at osf.io/gvc2w/ and can be used to address questions about semantic and syntactic composition.

CORRESPONDING AUTHOR:
**Arianna Zuanazzi**

Department of Psychology, New York University, New York, USA; Center for Language, Music and Emotion (CLaME), New York University, New York, USA

az1864@nyu.edu

KEYWORDS:
compounds; thematic roles; diathesis; semantics; syntax

TO CITE THIS ARTICLE:

# (1) BACKGROUND

While the meaning of lexicalized compound words (e.g., "apple pie" or "spotlight") is known and stored in the mental lexicon, the meaning of novel compounds (e.g., "doctor charity") must be built online from the combination of the single constituents' concepts, in the absence of syntactic cues (such as verbs or prepositions; Gagné, 2001). In particular, English endocentric noun-noun compounds (NNCs) such as "doctor charity" are formed by combining two noun constituents: the first constituent (e.g., "doctor") is the modifier and the second constituent (e.g., "charity") is the head. Syntactic and semantic categories of NNCs are determined by the head (i.e., "doctor charity" is a "charity", not a "doctor"; Bauer, 1983; Spencer, 1991).

The mechanism through which NNC constituents are combined to derive new meanings has been matter of debate (see *Competition-Among-Relations-in-Nominals*, Gagné, 2000, 2001; Gagné & Shoben, 2002; *Schema-modification* theory, Murphy, 1988, 1990; *Dual-process* theory, Wisniewski, 1996). The overall picture that emerges from studies investigating novel NNC processing is that head and modifier differentially contribute to the formation of the underlying syntactic and semantic relations. The attempt to identify underlying relations has led to numerous taxonomies. For example, NNCs have been categorized based on predication, such as *make, use, causes, used by* (Levi, 1978; Shoben, 1991); based on semantic relations, such as *time, possessor+possessed, location* (Tratz & Hovy, 2010), or *topic, container, material* (Kim & Baldwin, 2005); and based on the linking prepositions, such as *about, from, for* (Ponkiya et al., 2018). Bisetto & Scalise (2005) suggested a simplified and cross-linguistically valid classification of compound words into three groups: *subordinate, attributive,* and *coordinate*.

Critically, neither traditional nor more recent NNC taxonomies outline an explicit and formal distinction between NNCs with grammatical relations in the active or passive voice, where head or modifier function as *agent*, respectively. However, thematic role assignment and verb diathesis play a fundamental role in sentence processing, thus possibly also affecting the processing of compounds. In support of this hypothesis, Ferretti & Gagné (2006) found that English speakers assign the grammatical role of *subject* to compounds' modifiers/heads that are *good agents* (e.g., "detective") and the role of *object* to modifiers/heads that are *good patients* (e.g., "suspect"), suggesting that *agentivity* plays a critical role in grammatical structure formation and selection of verb diathesis in NNC interpretation.

To fill this taxonomy gap, here we created a dataset of 205 novel NNCs by manipulating differences in agentivity between modifier and head. In an online

study, participants classified NNC interpretations as either active or passive, which led to the creation of a dataset of 108 NNCs with distinct active and passive interpretations. Furthermore, to strengthen our taxonomic characterizations of novel NNCs into active and passive relations and illustrate the reuse potential of our dataset, we conducted a second online study using the dataset of 108 NNCs, where we tested processing differences (i.e., interpretation time differences) as a function of modifier's agentivity.

Previous studies on sentence processing show that passive sentences are harder to process than active sentences (i.e., lower accuracy and slower processing times; (Brookshire & Nicholas, 1980; Ferreira, 1994, 2003; Finocchiaro et al., 2015; Mack et al., 2013; Turner & Rommetveit, 1967). These processing differences are discussed in terms of thematic reanalysis that takes place only for passive sentences (e.g., "the suspect is investigated by the detective"), through revision of the initial thematic role assignment. As an example, "suspect" and "detective" are initially interpreted as *agent* and *patient/theme* (according to canonical *subject-to-agent* and *object-to-patient* mapping), and then remapped to *patient/theme* and *agent*, guided by verb diathesis, semantic features, and world knowledge (Ferreira, 1994, 2003; Finocchiaro et al., 2015; Mack et al., 2013).

In our second study, we asked whether the strategy observed for sentence processing may also be adopted when processing novel NNCs (e.g., "doctor charity"), where no explicit verb aids thematic role (re-)assignment. According to this hypothesis, NNCs would be initially interpreted with head as agent (e.g., "charity"), and then reanalyzed if semantic/distributional features suggest otherwise (i.e., "charity" is less likely to be an agent than "doctor"). This would lead to increased interpretation times (here measured as the time to *select* an interpretation) for NNCs where the modifier has higher agentivity than the head (i.e., passive interpretation) than vice-versa (i.e., active interpretation). Our results confirm this hypothesis and offer an example of how our dataset can be used in future studies.

We believe that this dataset would be helpful to linguistics, psycholinguistics, and cognitive neuroscience researchers who are interested in investigating how compound words are decomposed, stored, and combined to form new meanings.

# (2) METHODS

## 2.1 STUDY DESIGN

Data of both studies were collected online via the SONA and Prolific crowdsourcing platforms. Experiments were hosted on Pavlovia (pavlovia.org) and developed using jsPsych version 7.1.2 (De Leeuw, 2015). For both

studies, we employed a within-participant design with two different conditions: NNCs with modifiers with low agentivity (e.g., "coffee") and NNCs with modifiers with high agentivity (e.g., "doctor"). Heads were set to have medium agentivity (e.g., "charity"). Agentivity was measured as the likelihood of a word being agent over the likelihood of being patient/theme in a sentence.

## Study 1: interpretation

Each participant was presented with 205 NNCs and was required to choose the interpretation that best matched the meaning of each NNC, between two possible interpretations. E.g., "doctor charity": (1) active interpretation: e.g., "a charity that focuses on doctors"; (2) passive interpretation: e.g., "a charity that is managed by doctors"; we also added (3) no interpretation: "this compound does not have a sensible interpretation"; and (4) other interpretation: "other", in which case participants were asked to type their interpretation. In both active and passive interpretations, the head of the NNC functioned as the grammatical subject (according to formal definitions of the concept of *head;* Bauer, 1983; Spencer, 1991). Active/passive interpretations (i.e., "focuses on", "is managed by") were kept the same for all NNCs to make multiple choices predictable. The "no interpretation" option was included to allow participants to indicate that they could not assign an interpretation to the NNC. The "other" option was included to avoid forcing a specific interpretation when not semantically suitable (i.e., "focuses on", "is managed by"). The order of stimuli was randomized within each participant. The order of choices of interest (1 and 2) was randomly alternated trial-by-trial. We collected participants' choices and trials where participants chose "no sensible interpretation" or "other" (8.5%) were excluded from further analysis. The experiment duration was approximately 30 minutes. Interpretation data are available at osf.io/gvc2w/.

To select NNCs with low-medium and high-medium agentivity that were more likely interpreted as active and passive, for each compound we performed binomial tests separately for active and passive interpretations and separately for head and modifier. We retained only constituent nouns with significant results in >60% compounds in which they were modifier or head, in

the predicted direction (i.e., active: low-medium; passive: high-medium). We then combined all the retained modifiers with high/low agentivity with all the retained medium-agentivity heads. This resulted in 108 NNCs with distinct active and passive interpretations, such as: 54 NNCs low-medium agentivity and active interpretation, and 54 NNCs with high-medium agentivity and passive interpretation. The dataset is available at osf.io/gvc2w/.

## Study 2: interpretation time

In study 2, we tested the 108 NNCs selected for active/passive interpretations in study 1. In each trial, participants were presented with a fixation cross for 1 s, followed by the two words of the NNC (e.g., "doctor charity") for 2 s. Words were presented on each half of the screen, 140 pixels from the fixation cross (Figure 1). This time allowed participants to read the NNC before selecting an interpretation. Participants were then required to press a "continue" button horizontally located in the center of the screen, which forced them to place their mouse cursor in a spatially neutral position, equidistantly between the first two choices (i.e., the choices of interest; Figure 1). Participants were then presented with the same four possible choices of study 1: (1) active interpretation: e.g., "a charity that focuses on doctors"; (2) passive interpretation: e.g., "a charity that is managed by doctors"; (3) "this compound does not have a sensible interpretation"; (4) "other", in which case they were asked to type their interpretation. Participants were asked to choose the interpretation that best matched the meaning of each NNC, using the mouse (Figure 1). The order of stimuli was randomized within each participant. The order of the choices of interest (1 and 2) was randomly swapped trial-by-trial. We collected participants' choice and interpretation time. The experiment duration was 20–30 minutes.

Trials where participants chose "no sensible interpretation" or "other" were excluded from further analyses (10.5%). Participants' interpretation time was computed as the time between pressing "continue" (i.e., repositioning the mouse cursor) and clicking on the preferred interpretation. We removed trials where interpretation time was >2 SD from the mean within

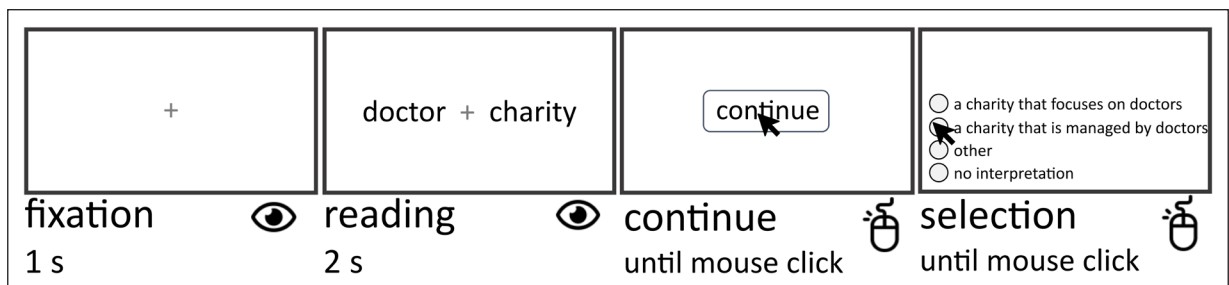

**Figure 1** Experimental procedure. Example trial procedure for the NNC "doctor charity". Participants read the novel NNC for 2 seconds and pressed "continue" to position the mouse cursor equidistantly between choices 1 and 2. Interpretation time was computed as the time between clicking "continue" and clicking on the preferred interpretation. The black arrow represents the mouse cursor.

each participant (4.5%). A square root transformation was applied to interpretation time to normalize the distribution. Interpretation time data are available at osf. io/gvc2w/.

To test whether interpretation time differed based on agentivity differences between modifier and head (associated to active/passive interpretations), we fitted a Linear Mixed-effects Model to the interpretation time data, with agentivity manipulation, constituent frequencies, the sum of the two constituents' bigram frequency, word length, and semantic similarity between the two constituents as fixed effects, and participants, modifier word, and head word as random intercepts. Analyses were conducted using the lme4 R package (Bates et al., 2015).

Our results show that participants spent significantly more time choosing the interpretation of NNCs with modifiers with high agentivity than low agentivity (Beta = 1.82, CI = [0.25, 3.39], $p$ = 0.046). This result demonstrates that processing cost of NNCs reflects agentivity differences between modifier and head. Participants were also significantly faster for NNCs with higher semantic similarity (Beta = −0.72, CI = [−1.11, −0.34], $p$ < 0.001). None of the other features (modifier frequency, head frequency, word length, and sum bigram frequency) showed significant effects on interpretation time ($p$ > 0.05).

## 2.2 TIME OF DATA COLLECTION

Data collection was performed between June 2021 and August 2021.

## 2.3 LOCATION OF DATA COLLECTION

Data acquisition was performed online. Participants were all located in the United States of America.

## 2.4 SAMPLING, SAMPLE AND DATA COLLECTION

### Participants of study 1

A group of 58 participants completed study 1 (i.e., interpretation). 26 participants (15 female; $M_{age}$ = 20.1 years; $SD_{age}$ = 1.90 years) were recruited via the platform SONA at New York University and were granted university credits for taking part in the study; 32 participants (16 female; $M_{age}$ = 32.9 years, $SD_{age}$ = 4.86 years) were recruited from the Prolific online crowdsourcing platform (prolific.co) and received monetary compensation for taking part in the study.

### Participants of study 2

A new group of 77 participants (59 female; $M_{age}$ = 19.36 years; $SD_{age}$ = 1.19 years) completed study 2 (i.e., interpretation time). They were recruited via the platform SONA at New York University and were granted university credits for taking part in the study. 5 participants

failed the attention checks (accuracy < 70%) and were not included in the final analysis, which included 68 participants.

All participants from both studies were English native speakers located in the United States of America, and self-reported normal hearing and normal or corrected to normal vision.

## 2.5 MATERIALS/SURVEY INSTRUMENTS

The linguistic stimulus set before performing study 1 included 205 novel NNCs. Novel NNCs were built as follows: (1) From the Glasgow Norms (Scott et al., 2019) and the University of South Florida Free Association Norms (Nelson et al., 2004) corpora, we selected nouns with similar bigram frequencies, word frequency, and number of syllables (2 to 3, 45 words); (2) we computed the likelihood of a noun being agent over the likelihood of being patient/theme in a sentence: frequencies of agent and patient/theme thematic roles were extracted from the Proposition Bank (Palmer et al., 2005), OntoNotes (Weischedel et al., 2011), and the Abstract Meaning Representation dataset (Banarescu et al., 2013), (3) We divided nouns into high (>1.5, 15 words), medium (>0.5 and < 1.5, 13 words) and low agentivity (< 0.5, 17 words); (4) we paired disyllabic and trisyllabic nouns with low agentivity in the modifier position with disyllabic and trisyllabic nouns with medium agentivity in the head position, respectively. The dataset is available at osf. io/gvc2w/. Informed by the results of study 1, we sub-selected a dataset of 108 NNCs with distinct active and passive interpretations: 54 with modifier-head with low-medium agentivity (active) and 54 with high-medium agentivity (passive).

Each of the unique words in the NNC datasets of study 1 and 2 is accompanied by the following lexical-semantic features of interest: number of letters, log-transformed word frequency (i.e., Log_Freq_HAL from the English Lexicon Project; Balota et al., 2007), bigram frequency, and semantic similarity between each two constituent words (i.e., extracted using cosine similarity of word2vec representations; Mikolov et al., 2013).

## 2.6 QUALITY CONTROL

Before performing study 2, we ran a pilot study with a different group of 26 participants (19 female; $M_{age}$ = 19 years, $SD_{age}$ = 1.04 years), to perform an a-priori power analysis. The a-priori power analysis (matched pairs, two-tails) on interpretation time differences between agentivity conditions (low-medium vs high-medium) indicated that a total of 36 subjects are required to achieve a power of 0.95 (G*Power3; Faul et al., 2009). We increased the sample size to account for the exclusion rate reported for online crowdsourcing experiments (Peer et al., 2017; Simcox & Fiez, 2014).

Throughout study 2, we also included 6 to 8 attention checks, to verify that participants were following the instructions. During these attention checks, participants were told to select one specific option (e.g., "select option 2"). Participants who failed the attention checks (accuracy < 70%) were excluded from the dataset.

## 2.7 DATA ANONYMISATION AND ETHICAL ISSUES

All participants provided written informed consent, as approved by the local institutional review board (New York University's Committee on Activities Involving Human Subjects). All data were anonymized for sharing purposes in public data repositories.

## 2.8 EXISTING USE OF DATA

The dataset of 108 NNCs was used in the interpretation time study described here (study 2) and cited in a manuscript submitted for peer-review.

## (3) DATASET DESCRIPTION AND ACCESS

### 3.1 REPOSITORY LOCATION

NNC stimuli (study 1 and 2), interpretation data of study 1 and 2, and interpretation time data of study 2 are available at osf.io/gvc2w/.

### 3.2 OBJECT/FILE NAME

In stimuli folder:
study1_stimuli_novel_compound_diathesis.csv
study2_stimuli_novel_compound_diathesis.csv

In data folder:
study1_data_novel_compound_diathesis.csv
study2_data_novel_compound_diathesis.csv

### 3.3 DATA TYPE

Stimuli: primary data
Data: processed data

### 3.4 FORMAT NAMES AND VERSIONS

Stimuli and Data: CSV
README: TXT

### 3.5 LANGUAGE

English

### 3.6 LICENSE

The data were published under a CC-BY Attribution 4.0 International (CC-BY 4.0) license.

### 3.7 LIMITS TO SHARING

N/A

### 3.8 PUBLICATION DATE

The dataset was published on OSF on April 27, 2023.

### 3.9 FAIR DATA/CODEBOOK

NNC datasets were published on the OSF platform, which ensures data findability and accessibility. We provided datasets in .csv format and accompanied by README files, to ensure data accessibility and reusability.

## (4) REUSE POTENTIAL

We created a large dataset of novel noun-noun compound words (NNCs), where modifier and head were chosen based on agentivity (i.e., the likelihood of being agent over the likelihood of being patient/theme in a sentence). We classified NNCs based on the verb diathesis of interpretations (study 1) and collected participants' interpretation time (study 2) to demonstrate that agentivity differences between modifier and head lead to measurable processing effects that are comparable to that found in sentence processing. Furthermore, we annotated every word with lexical-semantic features that have been shown to play a role during reading and listening, and are reflected in eye-tracking and neural measurements (Caffarra et al., 2021; Clifton et al., 2007; Frank & Willems, 2017).

Previous theoretical work and psycholinguistics research have classified compound words using numerous different taxonomies of relations. These taxonomies do not explicitly take into account the verb diathesis of the syntactic relation (active vs passive) which reflects the thematic structure of modifier and head (Levi, 1978; Shoben, 1991; Bisetto and Scalise, 2005; Kim & Baldwin, 2005; Ponkiya et al., 2018; Tratz & Hovy, 2010). The likelihood of a word being agent in sentences has been shown to play an important role in thematic role assignment in the case of novel compounds (Ferretti & Gagné, 2006). Our study aims to fill this classification gap by providing a dataset of 205 English novel NNCs with active and passive interpretation as a function of modifier-head agentivity. Furthermore, we made the interpretation time data collected for the 108 NNCs of study 2 available, to provide an example of how our dataset can be employed. We believe that this dataset could be helpful to linguistics, psycholinguistics, and cognitive neuroscience researchers who are interested in investigating how compound words are decomposed, stored, and combined to form new meanings. Furthermore, this dataset could be of interest to researchers investigating the creative potential of human language (e.g., Kuznetsova et al., 2013).

In particular, we showed that our dataset is suitable for online studies that aim to collect participants' choices, eye-movements, and time-resolved behavioral data (e.g.,

study 2). Online studies have become common practice in psycholinguistics research (Keuleers & Balota, 2015) and in cognitive neuroscience research more broadly (Johnson et al., 2022). Through crowdsourcing, large quantity of data are collected in a short amount of time (Simcox & Fiez, 2014). We showed that our dataset is suitable for a 20-30 minutes online study.

The datasets of study 1 and 2 contain NNCs created by combining a set of unique words, controlled for bigram frequencies, word frequency, and number of syllables. This carefully designed datasets are suitable for neuroimaging studies that aim to limit the amount of inter-trial variability to only manipulate the dimension of interest (Friederici, 2011).

Finally, given the surge of interest in large language models' behavior (Warstadt & Bowman, 2022), we envision this dataset being used by researchers interested in comparing how humans and language models derive syntactic structure and create new meanings.

## FUNDING INFORMATION

This work was supported by the Ernst Struengmann Foundation (DP) and the Leon Levy Foundation (AZ).

## COMPETING INTERESTS

The authors have no competing interests to declare.

## AUTHOR CONTRIBUTIONS

PC: study design, data collection, data analysis, data curation, manuscript writing and editing
DP: project supervision, manuscript editing
AZ: project supervision, study conceptualization, study design, data collection, data analysis, data curation, manuscript writing and editing

## AUTHOR AFFILIATIONS

**Phoebe Chen** orcid.org/0000-0002-9208-1682
Department of Psychology, New York University, New York, NY, USA

**David Poeppel** orcid.org/0000-0003-0184-163X
Department of Psychology, New York University, New York, NY, USA; Center for Language, Music and Emotion (CLaME), New York University, New York, NY, USA; Ernst Strüngmann Institute for Neuroscience, Frankfurt, DE

**Arianna Zuanazzi** orcid.org/0000-0002-8330-4450
Department of Psychology, New York University, New York, NY, USA; Center for Language, Music and Emotion (CLaME), New York University, New York, NY, USA

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

## PEER REVIEW COMMENTS

*Journal of Open Psychology Data* has blind peer review, which is unblinded upon article acceptance. The editorial history of this article can be downloaded here:

- **PR File 1.** Peer Review History. DOI: https://doi.org/10.5334/jopd.93.pr1

**TO CITE THIS ARTICLE:**

**Submitted:** 04 May 2023     **Accepted:** 05 July 2023     **Published:** 29 August 2023

