## [Peer Review History. · Journal of Open Psychology Data]

A dataset of 108 novel noun-noun compound words with active and passive interpretation

Phoebe Chen, David Poeppel, Arianna Zuanazzi

Response to Reviewers

We would like to thank the reviewers for their constructive and helpful comments and suggestions. We really appreciate the time and effort they spent on our manuscript.

Following the reviewers' comments, we have updated the manuscript with the required information, uploaded stimuli and data for "Study 1: interpretation" to OSF, and edited the dataset and README for "Study 2: interpretation time", according to the reviewers' suggestions. We have also reorganized the repository to be more easily browsable.

In the following, we present point-by-point responses, each followed by lines and page numbers of the changes in the manuscript. In addition, we have indicated the changes in orange in the manuscript.

REVIEW 1:

** Comments on the manuscript*

The manuscript is very clearly written at an appropriate level of detail. Below are a couple of minor comments:

2.1 Study design - it would be nice to provide a couple of concrete examples of NNCs with low and high agentivity.

Thank you, we have added examples for nouns with low, middle, and high agentivity (p.3, lines 38-39).

Study 1 - how many trials with "no interpretation"/"other" were excluded from the analysis?

We have added to the text that 8.5% trials were excluded from further analysis (p.4, line 18).

Study 2 - Fig. 1 seemed to suggest that the cursor is a bit closer to option 2 at the outset, but maybe the image is rendering a bit weirdly on my computer. Perhaps the notion of "spatially neutral" can be more clearly explained...

Thank you for the suggestion, we have now clarified the meaning of "spatially neutral": "which forced them to place their mouse cursor in a spatially neutral position, equidistantly between the

first two choices” (p.4, lines 39-40, and p.5, line 12). We acknowledge that the rendering of the procedure might show only an approximation of the real setup, due to space constraints. We have slightly edited the procedure figure to show the neutral position of the cursor more clearly.

Analysis of study 2 results - please clarify if the random effects are that of random intercepts or random slopes, and I would suggest citing the lme4 package to give credit to the package authors (<https://ropensci.org/blog/2021/11/16/how-to-cite-r-and-r-packages/>).

Thank you, we have now specified in text that they are random intercepts and cited the lme4 package (p.5, line 30-31).

2.5 Materials - I don't believe the abbreviation ELP was previously explained in the ms. In addition, it would be good to specify which frequency value was obtained from the ELP as it contains various types of frequency values from different corpora.

Thank you, we have added the full name of ELP and specified that we used the Log_Freq_HAL value (p. 7, line 7).

** Comments on stimuli_novel_compound_diathesis.csv*

I would suggest including the number of letters and letter bigram frequency for the head and modifier separately.

Thank you, we have added the number of letters and letter bigram frequency for each noun separately, and changed the README accordingly.

In the corresponding README file, the label for "high" is actually "hi" in the .csv file, and it would be good to clarify that these are /log/ frequency values.

Thank you, we have now replaced “hi” with “high” and clarified in the README that these are log frequency values.

In addition, I found it somewhat confusing that this file was said to contain the NNCs used in study 1 and 2 since there were only 108 NNCs. Didn't study 1 include over 200 NNCs? It would be useful to include the full dataset from study 1 since it can inform future researchers on how the NNCs were curated or be interesting to researchers who are looking for "ambiguous" NNCs.

Thank you for the helpful suggestion, we have now uploaded the complete dataset for Study 1: interpretation, which includes all 205 NNCs and their features (study1_stimuli_novel_compound_diathesis.csv) and all participants' choices (study1_data_novel_compound_diathesis.csv). We have updated the README accordingly and updated the manuscript's abstract.

Finally, a suggestion to highlight that choices that corresponded to no interpretation or other were excluded from this dataset - although such responses can still be useful - e.g., if someone wishes to conduct a qualitative analysis of alternative interpretations of various NNCs.

Thank you, we have included these responses in the `study1_data_novel_compound_diathesis.csv` dataset for Study 1, and added to the README that these were removed from the dataset of study 2.

Signed,
Cynthia Siew

REVIEW 2:

The paper describes very well how the dataset was created. The deposited data also has most of the details. I only have the following minor suggestions. Among these (2) and (3) are necessary, and (1) and (4) will make the dataset more useful to other researchers.

1. I couldn't understand the reason behind reporting "Study 1: interpretation" but not including the corresponding data with participants' choice responses. Since this data was the basis for selecting stimuli (that fulfill certain statistical criteria) for the next study, having this data is useful; without that the reader (user of the data) has to rely on authors' statistical analysis (not included) for selecting stimuli for Study 2. Perhaps authors could mention the reason in the paper (or include the data, and with all four types of responses).

Thank you for the helpful suggestion, we have now uploaded the complete dataset for Study 1: interpretation, which includes all 205 NNCs and their features (`study1_stimuli_novel_compound_diathesis.csv`) and all participants' choices (`study1_data_novel_compound_diathesis.csv`). We have updated the README accordingly and updated the manuscript's abstract.

2. In 'data_novel_compound_diathesis.csv' and the corresponding README:
- The column 'participant_anon' from the CSV is not defined in the README
- The column 'Agentivity' defined in the README but is not included in the CSV

3. In 'stimuli_novel_compound_diathesis.csv' and the corresponding README:
- The column 'Agentivity' defined in the README but is not included in the CSV

Thank you, we have now included "participant_anon" in the README and defined the agentivity column in the CSV and README.

4. In "Study 2: interpretation time", maybe the trials that were removed with the following criteria can be kept in; they could be informative to other researchers:

- "Trials where participants chose "no sensible interpretation" or "other" were excluded from further analyses (10.5%)"

- "We removed trials where interpretation time was > 2 SD from the mean within each participant (4.5%)"

Thank you, we have included the responses for "no sensible interpretation" and "other" in the `study1_data_novel_compound_diathesis.csv` dataset for Study 1. We have added to the README that these were removed from the dataset of study 2 and clarified that this dataset includes all interpretation times, before removing trials where interpretation time is > 2 SD from the mean.